# Negative Pressure Wound Therapy in the Treatment of Complicated Wounds of the Foot and Lower Limb in Diabetic Patients: A Retrospective Case Series

**DOI:** 10.3390/jcm14207193

**Published:** 2025-10-12

**Authors:** Octavian Mihalache, Laurentiu Simion, Horia Doran, Andra Bontea Bîrligea, Dan Cristian Luca, Elena Chitoran, Florin Bobircă, Petronel Mustățea, Traian Pătrașcu

**Affiliations:** 1Medicine School, “Carol Davila” University of Medicine and Pharmacy, 050474 Bucharest, Romania; 2Surgical Department I, “Dr. I. Cantacuzino” Clinical Hospital, 030167 Bucharest, Romania; 3General Surgery and Surgical Oncology Department I, Bucharest Institute of Oncology “Prof. Dr. Alexandru Trestioreanu”, 022328 Bucharest, Romania

**Keywords:** amputation, diabetic foot, gangrene, negative pressure, NPWT

## Abstract

**Background**: Diabetes-related foot diseases represent a global health problem because of the associated complications, the risk of amputation, and the economic burden on health systems. Negative pressure wound therapy (NPWT) is a technique that uses sub-atmospheric pressure to help promote wound healing by reducing the inflammatory exudate while keeping the wound moist, inhibiting bacterial growth, and promoting the formation of granulation tissue. **Objective**: This study aimed to assess the effectiveness of NPWT in preventing major amputation in diabetic patients with complicated foot or lower limb infections and to contextualize the results through a review of the existing literature. **Materials and methods**: We conducted a retrospective study at the First Surgical Department of “Dr. I. Cantacuzino” Clinical Hospital in Bucharest, Romania, over a 15-year period, including 30 consecutive adult patients with diabetes and soft tissue foot or lower limb infections treated with NPWT. Patients with non-diabetic ulcers, incomplete medical data, or aged under 18 were excluded. All patients underwent initial surgical debridement, minor amputation, or drainage procedures, followed by the application of NPWT using a standard protocol. Dressings were changed every 2–4 days for a total of 7–10 days. Antibiotic therapy was adapted according to the culture results. The primary outcome was limb preservation, defined as avoidance of major amputation. Secondary outcomes included in-hospital mortality and wound status at discharge. **Results**: NPWT was associated with a favorable outcome in 24 patients (80%), defined by wound granulation or healing without the need for major amputation. Five patients (16.6%) underwent major amputation because of failure of the primary lesion treatment, and one patient died. No statistically significant association was observed between the outcomes and standard classification scores (WIFI, IWGDF, and TPI). A comprehensive literature review helped to integrate these findings into the existing pool of knowledge. **Conclusions**: NPWT may support limb preservation in selected diabetic foot cases. While the retrospective design and the small sample size of the study limit generalizability, these results reinforce the need for further controlled studies to evaluate NPWT in real-life clinical settings. The correct use of NPWT combined with etiological treatment may offer a maximum chance to avoid major amputation in patients with diabetes-related foot diseases.

## 1. Introduction

The global diabetes mellitus prevalence is high and rising; in 2024, there were about 589 million diabetic patients (20–79 years), representing 1 in 9 people, and the number is expected to reach 853 million by 2050 [1]. According to the International Working Group on the Diabetic Foot, diabetes-related foot diseases are defined as when a person currently or previously diagnosed with diabetes mellitus presents one or more of the following: peripheral neuropathy, peripheral arterial disease (PAD), infection, ulcer(s), neuro-osteoarthropathy, gangrene, or amputation [2]. Patients with diabetic foot diseases have high disability and mortality rates, and the diabetic foot is considered one of the main health-related killers. Lifetime incidence of foot ulceration ranges from 19 to 34%, with a recurrence rate of 40% in one year and 65% in 3 years [3]. Diabetes-related foot diseases are also responsible for 75% of major amputations of the lower limb. Worldwide, an amputation due to a diabetes-related foot disease is performed every 20 s, and 70% of these patients are expected to die in the next 5 years, resulting in a mortality rate higher than those of most malignant tumors. Furthermore, the economic impact of diabetes-related foot diseases is substantial; annually, billions of dollars are needed for treatment of these patients.

Diabetic foot lesions need a multidisciplinary approach to treatment, the aim of which is to promote wound healing through infection control and prevention of recurrence to avoid major amputation. Besides glycemic control, off-loading and revascularization as a local intervention is often necessary, which may include debridement, drainage of abscesses, management of phlegmons, and even minor amputations [4]. These methods are considered standard wound care, but the persistent challenge in achieving timely wound healing necessitates finding new modalities of wound care.

Negative pressure wound therapy (NPWT) is a technique that uses sub-atmospheric pressure to help promote and optimize wound healing in diabetic foot; it has also been applied in the treatment of a large variety of acute and chronic lesions (e.g., dehiscent wounds, venous or neuropathic ulcers, necrotic fasciitis) [5]. The main mechanisms of NPWT involve reducing the inflammatory exudate while keeping the wound moist, inhibiting bacterial growth, and promoting the formation of granulation tissue [6,7]. Its advantages include improving wound blood perfusion; promoting cell proliferation, angiogenesis, and wound tissue repair; and regulating the signaling pathway to modulate cytokine expression [8].

Ever since its introduction in 1993 by the German physician Fleischmann, the advantages of NPWT have been recognized. Today, this type of therapy is used practically in all surgical specialties, both for the treatment of chronic or difficult wounds as well as for acute closed wounds. Diabetic foot diseases are not an exception; European and American guidelines recommend NPWT for the treatment of diabetic foot ulcers (DFUs) [9].

Despite the 2023 IWGDF guidelines [10] advising against the routine use of NPWT in infected diabetic foot ulcers, the clinical decision to initiate NPWT in selected septic cases within our cohort was based on real-life surgical judgment, multidisciplinary assessment, and the lack of alternative wound management strategies for advanced soft tissue infections. In these patients, infection control had been partially achieved through urgent surgical debridement, drainage, or minor amputation, and NPWT was introduced as an adjunctive measure to promote granulation and wound stabilization [5]. Several previous studies, although heterogeneous, have reported potential benefits of NPWT even in infected settings when combined with appropriate surgical and antibiotic management. Our approach reflects the complexity of treating diabetic foot infections in practice, where guidelines may not always accommodate the full clinical spectrum of tissue loss, systemic sepsis, or wound evolution.

Given the ongoing debate and conflicting evidence regarding the use of negative pressure wound therapy in diabetic foot management, particularly in cases complicated by infection, the aim of this study was to retrospectively evaluate the role of NPWT in a series of diabetic patients with complex foot or lower limb wounds that were managed in a real-life tertiary care setting. We sought to determine whether NPWT contributed to limb preservation and to analyze outcomes in relation to established classification systems. Additionally, we aimed to contextualize our findings within the current body of literature, including recent guideline updates, and to explore the potential benefits and limitations of NPWT in this patient population. The study’s primary outcome was major amputation avoidance, and the secondary outcomes were wound healing, complication rates, and mortality.

## 2. Materials and Methods

This study was conducted in accordance with the Declaration of Helsinki and approved by the Institutional Ethics Committee of “Dr. I. Cantacuzino” Clinical Hospital, Bucharest (protocol code 14918 and 15 July 2025).

The design of the study was a retrospective case series and entailed the observation of all adult patients with diabetic foot-related soft tissue infections treated with NPWT over a 15-year period (January 2008–December 2023). Although limited, the number of patients represented the totality of patients treated by a single team in the Department of Surgery of “Dr. I Cantacuzino” Clinical Hospital—this approach ensured a uniform application of intervention and follow-up plan.

The exclusion criteria were age under 18 years, refusal to sign informed consent for anonymous medical data collection for medical studies, incomplete clinical data, being a non-diabetic patient, the presence of ulcers/ischemic gangrene requiring primary major amputation, and non-infectious/non-complicated wounds.

The patients’ data was extracted from the hospital’s database and from patient records. All patients signed an informed consent for the medical procedure, including the proposed treatment and follow-up plan. They also gave consent for the use of aggregated results and medical data (anonymously) for the purpose of elaborating and publishing medical papers.

The following variables were collected from patient records: demographic data (age and sex), diabetes type and duration, presence of comorbidities (peripheral arterial disease, chronic kidney disease, coronary artery disease, smoking status, and HbA1c levels), type and location of foot lesion, infection status (culture results and germ type), type of surgical intervention, classification scores (WIFI, IWGDF grade, and TPI), NPWT parameters (number of days, dressing changes, and pressure settings), and final outcome.

Negative pressure wound therapy (NPWT) was initiated following the primary surgical management—either a minor amputation, drainage of fluid collections, or aggressive wound debridement—once the wound bed had been cleaned and adequately prepared. The NPWT system used was the V.A.C.^®^ Therapy System (KCI/3M) in combination with a reticulate polyurethane foam (GranuFoam™), which has been in clinical use since the late 1990s and is specifically designed to enhance granulation tissue formation and wound exudate management [11]. The V.A.C.^®^ Therapy System was employed in all cases, using polyurethane foam dressings covered with an occlusive adhesive drape and connected to a portable negative pressure unit. Continuous negative pressure of −125 mmHg was applied, in accordance with standard protocols for infected or complex diabetic foot wounds. The treatment was applied for a limited duration of 7 to 12 days, during which three dressing changes were performed. The initial dressing was changed after 48 h, and subsequent changes were performed every 3 to 4 days, depending on wound characteristics and clinical evolution. Additional sharp debridement was performed at each dressing change when necessary. After the NPWT protocol was completed, the patients continued standard wound care according to their wound progression and local resources, including secondary closure or conservative outpatient management.

A favorable outcome was defined as wound granulation or healing at the moment of discharge, with limb preservation. Unfavorable outcomes included the need for major amputation (below or above knee) or in-hospital death. Patients discharged with a granulating wound were considered to have a favorable result, provided that the wound showed viable tissue and no surgical reintervention was needed.

Descriptive statistics (mean, standard deviation, and percentages) were used to summarize the characteristics of the patient cohort. Associations between the outcome categories and classification systems (WIFI, IWGDF grade, and TPI score) were assessed using Chi-square tests for categorical variables. A *p*-value < 0.05 was considered statistically significant. Statistical analyses were performed using SPSS software, version 23.0 (IBM Corp., Armonk, NY, USA).

## 3. Results

In Figure 1, we present the STROBE (Strengthening the Reporting of Observational Studies in Epidemiology) flow diagram for our study [12]. We identified 30 patients meeting the inclusion criteria; although this is a relatively small number of cases, they reflect the rarity with which NPWT is currently being applied in our tertiary facility. However, the fact that all cases were gathered from patients treated by the same team ensured a highly homogenous application of treatment interventions and follow-up procedures, thus reducing the overall bias of this retrospective cohort study. The indication for negative pressure therapy was large and difficult wounds associated with sepsis (24 cases) or mixed chronic lesions associated with ischemia (6 cases). The primary goal of this treatment was to avoid major amputation and prevent the loss of a limb.

Data related to the initial evaluation of the patients, details about how NWPT was applied, and the results obtained are summarized in Table 1.

Twenty-four patients had a favorable outcome and six had an unfavorable outcome. Of the latter patients, one died and five needed a major amputation. There were 27 patients with lesions of the foot, and 3 had extensive soft tissue infections of the lower limb.

A classification of the lesions was made according to the WIFI and IWGDF classification systems, and the Therapeutic Prognostic Index (TPI) [13] was also calculated. No association between the primary outcome and these classifications was found in this cohort (Table 2). In 26 out of 30 patients, the risk of amputation was high according to the WIFI classification, and 12 patients out of 30 were grade 4 according to the IWGDF classification. Ten patients had a TPI higher than 6, which indicates an increased probability of a major amputation. There were no associations found between the infection grade or TPI and the failure of NPWT to prevent a major amputation.

Clinically, the wounds differed in size, shape, and location, ranging from small wounds after a radius amputation to large foot and calf wounds after extensive infections (Figure 2).

There were 14 ray amputations performed, involving one to three toes, 4 transmetatarsal amputations, 11 debridement operations associated with fasciectomies and drainage, and 1 above-knee amputation with extensive debridement for gas gangrene to treat an extensive wound (Figure 3). These diverse medical procedures make comparison and standardization of treatment very difficult.

Germ isolation was performed and identified on cultures in 24 cases (Table 3), among whom two types of bacterial pathogens were identified in 7 cases.

The most frequently identified bacterial pathogen was Methicillin-resistant *Staphylococcus aureus* (MRSA), followed by *E. coli*. SA was also found in five out of the seven cases presenting with two pathogens. In six cases, the cultures were negative mainly due to previous antibiotic therapy. In three out of seven cases with infection caused by MRSA, the treatment failed to avoid a major amputation. Antibiotic therapy was initially empirical and, after the germ identification, modified accordingly based on the antibiogram and extended to the entire period of negative pressure wound therapy.

Among the 24 cases with a favorable outcome, only 6 patients were completely healed at the moment of discharge: 3 cases with skin graft and another 3 in whom the wound was sutured. The other 18 cases were discharged with a granulated wound and were scheduled for additional follow-ups.

For the five (16.6%) cases in whom a major amputation was needed to obtain healing, four had the amputation below the knee and one above the knee. Four of these patients had arteriopathy, and one patient was diagnosed with mixed neuro-ischemic diabetic foot.

## 4. Discussion

In our case series, NPWT was used as backup solution to avoid the need for a major amputation. Even though this goal was accomplished in 80% of cases, it is challenging to draw a definite conclusion based on these data only, so we also conducted a literature review.

The main therapeutic principles for diabetic foot wounds are control of the infection, improvement of local tissue perfusion, offloading, and promotion of tissue repair. NPWT has become an important asset in the therapeutic arsenal for the management of diabetic foot wounds due to its effects of enhancing local perfusion, promoting granulation tissue growth, and improving wound healing [14].

Given the clinical complexities and the variety of available treatment options for diabetic foot lesions, it is essential to generate robust evidence regarding the comparative effectiveness of NPWT versus standard wound treatment. Previous studies comparing NPWT and standard wound care for DFUs have reported mixed outcomes [15].

Since the late 1990s when NPWT became commercially available, many publications and basic studies have suggested the positive effects of NPWT on wound healing. By contrast, other series of studies and meta-analyses have found little evidence that NPWT provides better results than standard wound care.

Ove the past two decades, several studies have reported encouraging outcomes regarding the use of NPWT in patients with DFUs, particularly in complex or postoperative wounds. In our case series, 80% of patients avoided a major amputation after a limited course of NPWT, which is consistent with the published success rates ranging from 70% to 85% in selected cohorts. For example, a multicenter randomized controlled trial published in 2008 by Blume et al. (including the largest number of patients with chronic diabetic foot ulcers—342) found that patients treated with NPWT had significantly higher rates of wound closure and limb preservation compared with those receiving advanced moist wound therapy alone [16]. A complete ulcer closure was obtained in 43.2% of patients treated with NPWT versus 28.9% treated with moist dressings. Moreover, the amputation rates were significantly lower in the group of patients treated with NPWT compared with those treated with advanced moist therapy (4.1% vs. 10.2%), when both minor and major amputations were considered. Similarly, a prospective study conducted in 2005 by Armstrong and Lavery (including 162 patients from 18 centers with diabetic foot wounds who underwent partial amputation of the foot and were treated with NPWT until complete healing or up to a period of 16 weeks) demonstrated a better rate of complete healing, faster granulation and healing times, and fewer complications in NPWT-treated diabetic foot wounds [17]. These findings support our own observations, despite the differences in study design and sample size.

Notably, while many clinical trials have excluded infected or ischemic wounds, real-world case series have increasingly explored NPWT as an adjunctive tool in septic or borderline ischemic lesions. Our cohort reflects this clinical reality, in which NPWT was applied post-debridement to infected wounds under controlled hospital conditions. Although the 2023 IWGDF guidelines recommend against NPWT in active infections, previous observational studies [18,19] have shown that, when used judiciously after surgical infection control, NPWT may support healing and reduce tissue loss. Our results echo this pragmatic use of NPWT and suggest that a strict interpretation of current guidelines may not always capture the nuanced, multidisciplinary decision-making required in complex limb salvage situations.

A recent publication of the “German DiaFu RCT” compared NPWT with standard moist wound care (SMWC) for the treatment of diabetic foot ulcers in a real-life clinical practice. The study included all diabetic patients with foot lesions regardless of their neuropathic or angiopathic etiology and did not exclude patients with concomitant diseases that might negatively impact wound healing. Therapy application was performed at the discretion of the attending physician. This corresponds with the real-life situation of patients, so the results can be generalized and applied to current clinical practice. The German DiaFu RCT found that wound closure rate and time to complete wound closure were not significantly different between NPWT or SMWC. A large number of patients were lost to follow-up at the end of the study and missing endpoint documentations limited the validity of the analysis [20].

Two large meta-analyses showed that a large number of studies on NPWT had been published, but the evidence of its effectiveness is still low. The first meta-analysis was published in 2018 and included eleven RCTs with 972 participants. Nine of these studies examined patients with DFUs and the other two analyzed post-amputation wounds. Ten of these studies compared NPWT with wound dressing, and one compared the effect of NPWT at two different pressure settings. The authors found that the conclusions and results of the studies are imprecise and present risk of bias. Therefore, there is low-certainty evidence to suggest that NPWT may increase the proportion of wounds healed and reduce the time to healing for postoperative foot wounds and ulcers in people with DM [21].

The second meta-analysis screened almost 400 articles and included only 9 in the review, with a total of 943 patients, which was very similar to the sample size of the first meta-analysis. These studies were published in the last 10 years and presented a better description of the randomization method used. Wound healing rate, granulation tissue formation time, incidence of adverse reactions, and amputation rate were statistically analyzed. The results showed that NPWT could promote and accelerate wound healing, with similar rates of adverse events and amputations to conventional moist therapy [22].

Clinical data from existing RCTs and non-RCTs recommend the use of NPWT in the treatment of diabetic foot lesions, although the data obtained from these meta-analyses do not seem to specifically favor NPWT over standard treatment. At the same time, it is important to revise clinical practice guidelines for the diabetic foot regarding the use of NPWT [23].

The latest general diabetes guidelines published annually by the American Diabetes Association do not specifically mention NPWT but acknowledge it as a treatment option for diabetic foot ulcers. NPWT is recommended when wound infection is controlled, bleeding risk is managed, and ischemia is addressed [24,25].

In 2017, The European Wound Management Association published an extensive summary on the use of NPWT in different clinical situations, including for diabetic foot wounds. The guidelines suggest that complications such as ischemia and infection must be treated before applying NPWT. Technical progress in the development of NPWT devices in recent years is pointed out, and it is concluded that NPWT is an important adjuvant therapy in the management of DFUs and that its increased use in this field may be expected [26].

The International Working Group on the Diabetic Foot released the last guidelines in 2023. They recommend the use of NPWT as an adjunct therapy to standard of care for the healing of postsurgical diabetes-related foot wounds but advise against its use in non-surgical diabetes-related foot ulcers. In addition, it is not recommended for the treatment of diabetic foot-related infections [27,28].

Regarding classification systems, our study did not find a statistically significant association between the primary outcome and the WIFI, the IWGDF risk categories, or the Therapeutic Prognostic Index. This contrasts with some prior studies that reported predictive value for these systems in guiding treatment and estimating amputation risk. The discrepancy may be related to the small sample size, retrospective nature, and clinical heterogeneity of our cohort. Nevertheless, our experience suggests that NPWT may offer benefit even in cases deemed high risk by conventional scoring, reinforcing the need for individualized assessment beyond algorithmic thresholds.

### Study Limitations

This study has several limitations that must be acknowledged. First and foremost, the retrospective case series design inherently limits the ability to draw causal inferences. The absence of a control group and the lack of randomization expose the study to selection bias and the potential influence of confounding variables on both clinical decisions and outcomes. Although the cohort included all eligible patients treated with NPWT by the same surgical team over a 15-year period, the relatively small sample size (*n* = 30) and the long enrollment period raise concerns regarding the temporal consistency of patient selection, treatment protocols, and follow-up practices. Additionally, the monocentric nature of the study restricts the generalizability of our findings to other healthcare systems or institutions with differing standards of care and resource availability.

Another important limitation is the clinical heterogeneity of the cases included. Our patients presented with varying degrees of lesion severity, infection status, ischemia, and comorbidities, which complicates the comparison of outcomes and limits the strength of their associations with classification systems such as the WIFI, IWGDF, and Therapeutic Prognostic Index (TPI). Moreover, data on long-term outcomes, such as limb salvage beyond hospital discharge, wound recurrence, or patient-reported quality of life, were not available due to the lack of systematic outpatient follow-up. The study also relied on medical record documentation, which may have introduced information bias due to missing data or subjective clinical assessments. The retrospective nature of data collection precluded detailed recordings of certain variables that may influence healing outcomes, such as nutritional status, precise duration of diabetes, time to granulation tissue formation, and patient adherence, limiting the generalizability of the findings.

Additionally, while NPWT was applied using a standardized protocol, variations in individual wound characteristics and concurrent treatments could have affected the therapeutic response, further limiting the internal validity of the study.

An additional limitation lies in the contextual divergence between the clinical setting of this study and the ideal conditions recommended by current international guidelines. Notably, the 2023 IWGDF guidelines advise against the routine use of NPWT in infected diabetic foot ulcers due to insufficient high-quality evidence supporting its efficacy in such settings. However, our study reflects real-world clinical practice in a resource-constrained environment, where individualized decision-making often takes precedence over guideline-driven algorithms. This pragmatic approach, while clinically justified, limits the comparability of our results to those obtained in more structured, protocolized, or multicentric trials. As such, our findings may apply to similar tertiary surgical centers but are not generalizable more broadly.

## 5. Conclusions

Even though it is still challenging to obtain undeniable statistical evidence of the effectiveness of NPWT in treating diabetic foot lesions and avoiding major amputations, NPWT remains a valuable adjunctive tool for the treatment of these patients. The correct use of NPWT combined with etiological treatment may offer a maximum chance to avoid a major amputation and obtain wound healing in patients with diabetes-related foot diseases. Further high-quality RCTs are needed to clarify the exact role of NPWT in wound healing and wound area reduction—outcomes that are essential for the prevention of amputation in patients with diabetic foot lesions.

## Figures and Tables

**Figure 1 jcm-14-07193-f001:**
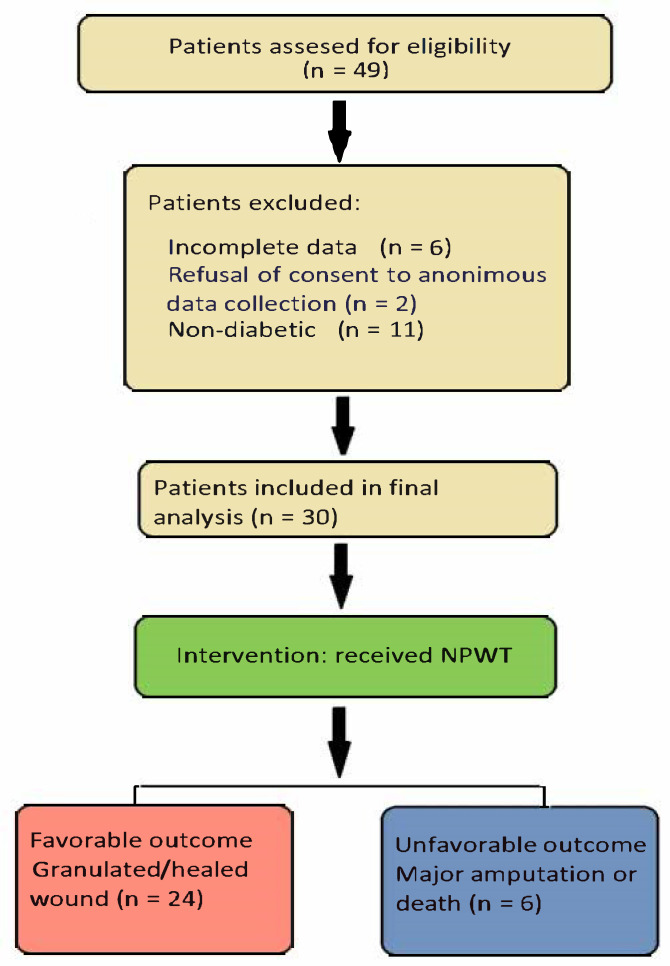
STROBE (Strengthening the Reporting of Observational Studies in Epidemiology) flow diagram.

**Figure 2 jcm-14-07193-f002:**
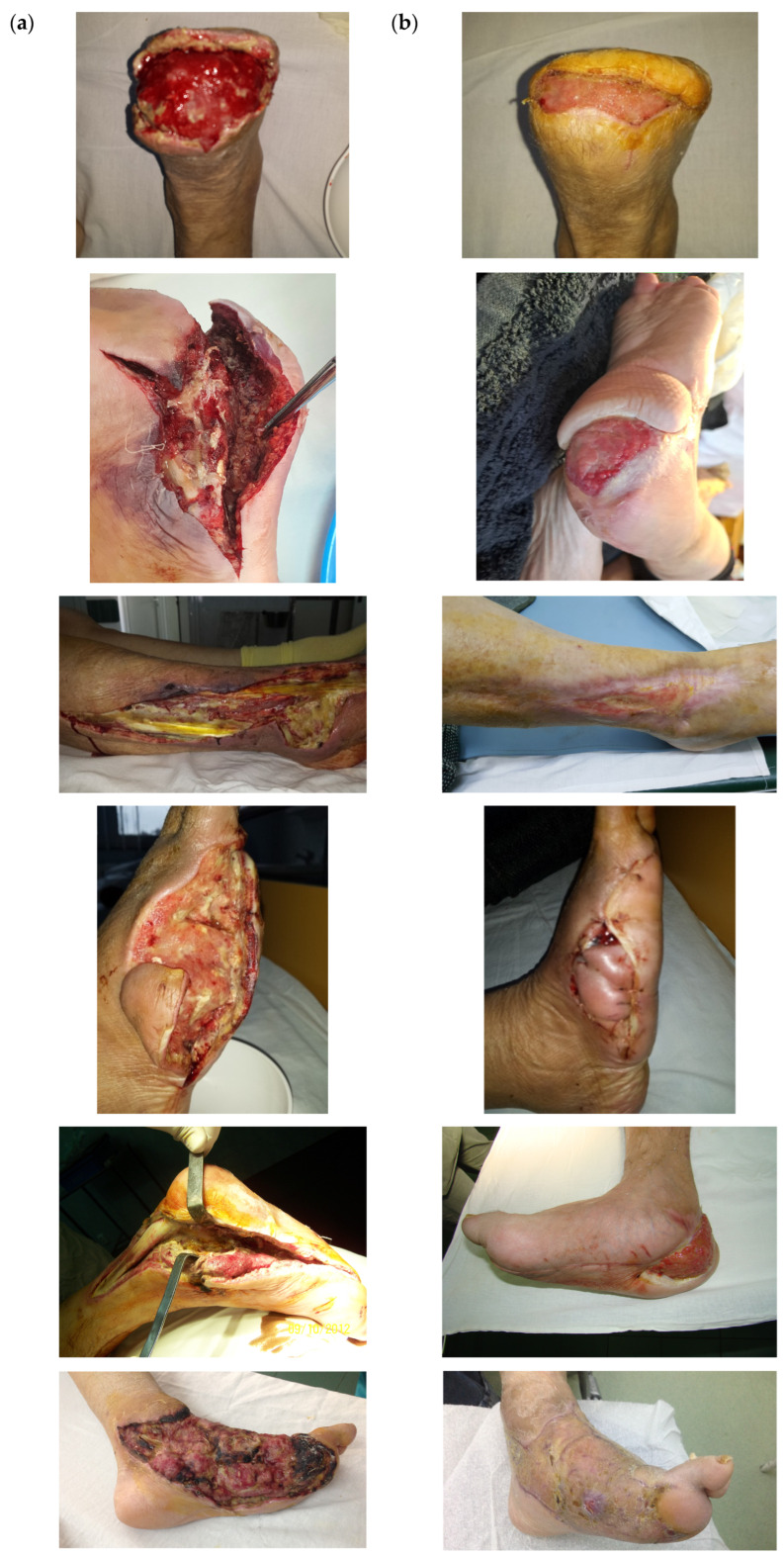
Clinical aspects of wounds treated with NPWT in 6 selected patients: (**a**) wound at the initiation of NPWT and (**b**) at 3 to 6 months after treatment.

**Figure 3 jcm-14-07193-f003:**
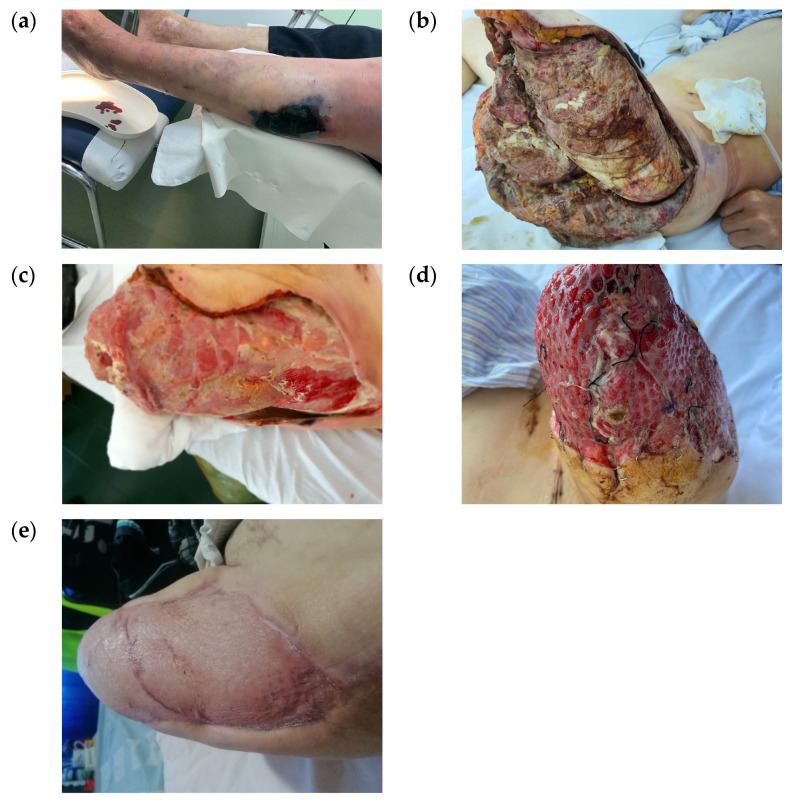
Exemplification of severity of wounds where NPWT was used case presentation: (**a**) a 42-year-old male with diabetes mellitus type I and history of multiple surgical interventions on both feet and chronic plantar ulcer was admitted for gas gangrene with septic shock (altered general condition, hypotension, severe anemia (6 g/dL), and leukocytosis (51,100/mm^3^)); (**b**) above-knee amputation was performed at admission; (**c**) NPWT was initiated on the second day and was continued for 2 weeks; (**d**) after the wound was covered with a skin graft; and (**e**) the final result at 3 months showing complete healing.

**Table 1 jcm-14-07193-t001:** Summary of cohort characteristics, treatment, and outcomes.

Variable	Value
Age	59.16 years (±10.32)
Sex	
Female	6 (20%)
Male	34 (80%)
Smoking	
Yes	14 (46.67%)
No	16 (53.33%)
Type of lesion	
Neuropathic	9 (30%)
Ischemic	14 (46.67%)
Neuro-ischemic	7 (23.33%)
Associated vascular disease	
Yes	21 (70%)
No	9 (30%)
Other diabetes-related conditions	
Retinopathies	3 (10%)
Renal impairment	6 (20%)
Mean duration of NPWT	9.1 days (±2.6)
Major amputation after the use of NPWT	5 (16.6%)
Healed or granulated wounds	24 (80%)
In-hospital mortality	1 (3.33%)
Mean hospitalization time	25.4 days (±10.4)

Abbreviation: NPWT—negative pressure wound therapy.

**Table 2 jcm-14-07193-t002:** Correlation between type and classification of lesions and probability of a major amputation after NPWT.

Variable	Value	Correlation Statistics
Type of lesion		Χ^2^ = 0.57 (*p*-value = 0.75)
Neuropathic	9 (30%)	
Ischemic	14 (46.67%)	
Neuro-ischemic	7 (23.33%)	
IWGDF classification		Χ^2^ = 1.2 (*p*-value = 0.54)
Grade 2	6 (20%)	
Grade 3	12 (40%)	
Grade 4	12 (40%)	
Therapeutic Prognosis Index (TPI)		Χ^2^ = 0.12 (*p*-value = 0.72)
<6	20 (66.67%)	
>6	10 (33.33%)	

Abbreviations: NWPT—negative wound pressure therapy; IGWDF—International Working Group on the Diabetic Foot; TPI—Therapeutic Prognostic Index.

**Table 3 jcm-14-07193-t003:** Bacterial pathogens isolated from cultures.

Bacteria	Number of Cases
MRSA	8
*E. coli*	7
*Pseudomonas aeruginosa*	3
*Enterobacter*	3
Coagulase-negative SA	3
*Klebsiella* spp.	2
Group B *Streptococcus*	1
Group D *Streptococcus*	1
*Proteus* spp.	1

Abbreviations: SA—*Staphylococcus aureus*; MR—Methicillin-resistant.

## Data Availability

The data supporting the findings of this study are not publicly available due to institutional restrictions. Requests to access these datasets will be considered upon reasonable request to the corresponding author H.D.

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
