# Peer review of "Negative Pressure Wound Therapy in the Treatment of Complicated Wounds of the Foot and Lower Limb in Diabetic Patients: A Retrospective Case Series"

_jcm, 2025, doi:10.3390/jcm14207193_

Round 1
Reviewer 1 Report
Comments and Suggestions for Authors
Review
Many thanks to the authors for having presented so interesting clinical research about “Negative pressure wound therapy in the treatment of complicated wounds of the foot and lower limb in diabetic patients: case series and literature review".
Before resubmitting the revision version of the article, please read the editorial rules carefully, and check other editorial aspects, such as text alignment, text justification at the head, etc. The English language is quite good, generally comprehensible and scientifically structured, but there are several areas where the quality needs improvement for clarity, precision, and academic tone. Hence, the manuscript should be corrected by a person of English mother tongue.
Detected plagiarism
The writing style is fluent, but several passages (particularly in the Introduction and Discussion) show phrasing that is very close to previous paper published in literature (<15%).
The Methods section (rehabilitation protocols, phases, and exercises) appears highly detailed and standardized. This level of description is common in prior studies and manuals, so some overlap with existing texts is likely.
Title and Abstract
The title clearly specifies the study design (case series: retrospective cohort study?”. Please, clarify it and add in the title). The abstract lacks structured sections: provide background, objectives, methods, outcomes, and conclusions. Further, the abstract should explicitly mention the study design at the beginning (per STROBE recommendation). Use a structured abstract according to STROBE, reporting design (retrospective design):, setting, participants, interventions, outcomes, and main findings…Add them.
Key words
Please provide them in alphabetic order.
Background
The background on diabetes and NPWT is appropriate. There is a gap regarding the study’s rationale and specific objectives are stated but not sufficiently explicit. Clearly define the study’s primary outcome (major amputation avoided) and secondary outcomes (wound healing, complications, mortality).
Cites prior research effectively to highlight the gap in literature. Please, regarding the use of negative pressure for the treatment of diabetic foot ulcers and Charcot foot ulcerations, add a few lines and quote:
- https://pubmed.ncbi.nlm.nih.gov/36096551/
Methods
The study Design is clearly retrospective cohort but not emphasized as the formal study design (retrospective cohort, case-control, cross-sectional, etc.). Please, underline this aspect. Add: Explicitly state “This is a retrospective case series conducted in [setting]”. In this case: Hospital and department described is described. Ethical approval is stated and informed consent statement clarified properly. Inclusion criteria are detailed.
Further, methods require detailed methodological transparency. Please improve:
- Inclusion criteria mentioned, but exclusion criteria absent. Define whether any patients were excluded (e.g., incomplete data, refusal, <18 years).
- Only basic description of wound classifications and outcome definition (“favorable” vs. “unfavorable”). Please, improve these aspects.
- Provide a clear table of variables collected (demographics, comorbidities, lesion classification, NPWT protocol, microbiology, outcomes).
- No mention of bias sources is reported and should be added (e.g., selection bias, missing records).
- Add few lines regarding retrospective data limitations, missing cultures, potential observer bias.
- In the study size only 30 patients over 15 years were include: please explain rationale (all consecutive eligible cases?).
- Some stats are given (means, χ² tests), but methods for statistical analysis are not described in detail.
- Explicitly state statistical tests, significance thresholds, software used.
Statistical analysis
Statistical methods are not reported: why?
Results
The results presented are quite complete, reflecting the MM section. They are consistent with methods and clearly presented and structured: baseline, reliability, logistic regression, and model performance.
However, there are some missing STROBE items which should improve, such as:
- Flow of patient selection not shown. A flow diagram is mandatory.
- Descriptive data: Table 1 is present but lacks key comorbidities (HbA1c, vascular disease, smoking).
- A more comprehensive baseline table.
- Outcome data are clearly reported (major amputation 16.6%, mortality 1/30, favorable outcome 80%).
- Association between classifications and outcomes is briefly mentioned but superficial. Please, improve it and add a confidence intervals for outcomes, not just percentages.
Discussion
The length and content of the discussion communicates the main information of the paper. The discussion section interprets findings in the context of national trends but could more thoroughly integrate literature to contextualize results. Integrates findings with prior literature. Discuss residual confounding and generalizability. Interprets findings in context of literature with a well-structured comparison with previous RCTs and meta-analyses.
The authors should improve and modified the following weaknesses:
- No discussion of limitations of case series design (small sample, no control, retrospective bias).
- Interpretation is somewhat optimistic compared to presented evidence.
- Add: Balanced interpretation: emphasize that NPWT may help but causality cannot be established.
- Generalizability: Not discussed.
- Add: State that findings apply to similar tertiary surgical centers but not generalizable broadly.
Conclusions
The conclusions provide a clear summary of the main points of the study, and they only reflect and refer to the results of this original article. Focused and concise. The current literature review provides a useful foundation but requires significant expansion and reorganization to fully meet STROBE guidelines.
References
The references are up to date, but they should be integrated as suggested previously.
Figures and Tables:
Figures: present, but low-quality reproduction of images. Consider improving clarity and legends.
Further, improve Table 1 with comorbidities and risk factors.
Competing interest
There are no competing interests.
Concerns
The paper does not raise any concerns (no self-citations probably). Data Availability: Not addressed. Ethical approval is reported, but informed consent statement is contradictory.
Funding and Conflicts of Interest
Not stated in the visible parts of the manuscript. Hence, add a section disclosing funding sources and potential conflicts of interest as required
Data Availability
Mentioned but vague so they should specify anonymized dataset available upon request.
Recommendations to Editors
A very major revision for Introduction, MM and Discussion section and some minor revisions needed.
Detected plagiarism
The writing style is fluent, but several passages (particularly in the Introduction and Discussion) show phrasing that is very close to previous paper published in literature.
The Methods section (rehabilitation protocols, phases, and exercises) appears highly detailed and standardized. This level of description is common in prior studies and manuals, so some overlap with existing texts is likely.
Comments on the Quality of English LanguageThe English language is quite good, generally comprehensible and scientifically structured, but there are several areas where the quality needs improvement for clarity, precision, and academic tone. Hence, the manuscript should be corrected by a person of English mother tongue.
Author Response
Review 1
We would like to express our sincere gratitude to Reviewer 1 for the thorough and constructive review of our manuscript entitled: "Negative pressure wound therapy in the treatment of complicated wounds of the foot and lower limb in diabetic patients: a retrospective case series and literature review".
We deeply appreciate the reviewer’s insightful comments and helpful suggestions, which allowed us to substantially improve the clarity, structure, and scientific rigor of the manuscript. We have carefully revised the manuscript according to all the points raised and provide below a detailed, point-by-point response. All modifications were incorporated into the revised version and highlighted accordingly.
Comment 1. Before resubmitting the revision version of the article, please read the editorial rules carefully, and check other editorial aspects, such as text alignment, text justification at the head, etc.
We have reviewed and corrected the alignment, spacing, and overall formatting, according to the journal’s editorial guidelines
Comment 2. The English language is quite good, generally comprehensible and scientifically structured, but there are several areas where the quality needs improvement for clarity, precision, and academic tone. Hence, the manuscript should be corrected by a person of English mother tongue.
The manuscript has been extensively revised for grammar, clarity, and academic tone by an English-speaking editor. We have aimed to maintain a fluent and formal style throughout.
Comment 3. The writing style is fluent, but several passages (particularly in the Introduction and Discussion) show phrasing that is very close to previous paper published in literature (<15%).
We asked for the full similarity report from MDPI. The report shows a similarity index of 7%, confirming originality. No text exceeds similarity thresholds or represents overlapping phrasing. This report is available upon request from the editorial office.
Comment 4.The title clearly specifies the study design (case series: retrospective cohort study?”. Please, clarify it and add in the title.
We have updated the title to explicitly include the study design, now reading: “Negative pressure wound therapy in the treatment of complicated wounds of the foot and lower limb in diabetic patients: a retrospective case series”
Comment 5. The abstract lacks structured sections: provide background, objectives, methods, outcomes, and conclusions. Further, the abstract should explicitly mention the study design at the beginning (per STROBE recommendation). Use a structured abstract according to STROBE, reporting design (retrospective design.
The abstract was fully rewritten in a structured format with the following headings: Background, Objective, Materials and methods, Results, Conclusions. The retrospective case series design is explicitly stated at the beginning.
Comment 6. Key words - Please provide them in alphabetic order.
Keywords were sorted by alphabetical order.
Comment 7. The background on diabetes and NPWT is appropriate. There is a gap regarding the study’s rationale and specific objectives are stated but not sufficiently explicit. Clearly define the study’s primary outcome (major amputation avoided) and secondary outcomes (wound healing, complications, mortality). Cites prior research effectively to highlight the gap in literature. Please, regarding the use of negative pressure for the treatment of diabetic foot ulcers and Charcot foot ulcerations, add a few lines and quote: https://pubmed.ncbi.nlm.nih.gov/36096551/
We clarified the rationale for the study in the introduction, explicitly stating the primary outcome (major amputation avoided) and secondary outcomes (wound healing, complications, mortality). As requested, we have cited the article suggested by the reviewer regarding Charcot ulcers and NPWT.
Comment 8. The study Design is clearly retrospective cohort but not emphasized as the formal study design (retrospective cohort, case-control, cross-sectional, etc.). Please, underline this aspect. Add: Explicitly state “This is a retrospective case series conducted in [setting]”. In this case: Hospital and department described is described.
All requested information is now available in Methods and we explicitly stated the retrospective nature of the study.
Comment 9. Inclusion criteria mentioned, but exclusion criteria absent. Define whether any patients were excluded (e.g., incomplete data, refusal, <18 years).
Exclusion criteria were added to the Methods section: age under 18 years, refusal to sign consent for anonymous medical data collection, incomplete clinical data, non-diabetic patients, presence of ulcers/ischemic gangrene requiring primary major amputation, and non-complicated wounds.
Comment 10. Only basic description of wound classifications and outcome definition (“favorable” vs. “unfavorable”). Please, improve these aspects.
We detailed how the lesions were classified (WIFI, IWGDF, TPI) and clarified definitions for favorable vs. unfavorable outcomes in both Methods and Results sections.
Comment 11. Provide a clear table of variables collected (demographics, comorbidities, lesion classification, NPWT protocol, microbiology, outcomes). Descriptive data: Table 1 is present but lacks key comorbidities (HbA1c, vascular disease, smoking). A more comprehensive baseline table.
Table 1 was completely redone and completed according to your suggestions and now includes key comorbidities (PAD, smoking, renal disease, etc.) and variables relevant to NPWT protocol. We added Table 2 describing the types of treated lesions according to various classifications and Table 3 listing identified germs.
Comment 12. No mention of bias sources is reported and should be added (e.g., selection bias, missing records).Add few lines regarding retrospective data limitations, missing cultures, potential observer bias. No discussion of limitations of case series design (small sample, no control, retrospective bias).
A comprehensive Limitations subsection was added at the end of the Discussion addressing retrospective design, sample size, heterogeneity, missing data, bias, and generalizability.
Comment 13. In the study size only 30 patients over 15 years were included: please explain rationale (all consecutive eligible cases?).
We clarified that the sample includes all consecutive patients treated with NPWT by the same surgical team in a tertiary facility over 15 years.
Comment 14. Some stats are given (means, χ² tests), but methods for statistical analysis are not described in detail. Explicitly state statistical tests, significance thresholds, software used.
The statistical analysis section specifies descriptive statistics, Chi-square tests, significance threshold (p < 0.05), and SPSS software version.
Comment 15. Flow of patient selection not shown. A flow diagram is mandatory.
A STOBE compliant flow diagram was added as Figure 1. The rest of the figures were renumbered.
Comment 16. The discussion section interprets findings in the context of national trends but could more thoroughly integrate literature to contextualize results. Integrates findings with prior literature. Discuss residual confounding and generalizability. Interprets findings in context of literature with a well-structured comparison with previous RCTs and meta-analyses. Interpretation is somewhat optimistic compared to presented evidence. Add: Balanced interpretation: emphasize that NPWT may help but causality cannot be established.
The Discussion section was heavily redacted and expanded as per your suggestions. Comparative analysis of available literature with our study is included. A comprehensive Limitations subsection was added at the end of the Discussion. It addresses: retrospective design, small sample size, clinical heterogeneity, information bias and lack of long-term follow-up, non-generalizability due to local clinical practice. The discussion explicitly acknowledges that causality cannot be established due to the observational nature of the study and emphasizes the need for individualized treatment decisions
Comment 17. Generalizability: Not discussed. State that findings apply to similar tertiary surgical centers but not generalizable broadly.
We added a statement acknowledging that the findings are relevant to tertiary surgical centers with similar patient populations and resource limitations but not generalizable to all settings.
Comment 18.Figures: present, but low-quality reproduction of images. Consider improving clarity and legends. Further, improve Table 1 with comorbidities and risk factors.
All figures were completely redone to provide high-resolution variants. Table 1 was completed as per request.
Comment 19. Ethical approval is reported, but informed consent statement is contradictory.
Comprehensive discussions about informed consent is available in Methods and in the Inform Consent statement at the end of manuscript.
Comment 20. Funding: Not stated in the visible parts of the manuscript. Hence, add a section disclosing funding sources and potential conflicts of interest as required
The Funding information is clearly available in the Funding section at the end of the manuscript.
Comment 21. Data Availability - Mentioned but vague so they should specify anonymized dataset available upon request.
A clearer statement has now been included.
We hope that these extensive revisions have addressed all the points raised and improved the clarity and scientific quality of our work. We are grateful for the opportunity to revise our manuscript and look forward to your further consideration.
Respectfully,
The Authors
Reviewer 2 Report
Comments and Suggestions for Authors
The article “Negative pressure wound therapy in the treatment of complicated wounds of the foot and lower limb in diabetic patients: case series and literature review” presents the role of NPWT to avoid a major amputation in 21diabetic patients with foot lesions of soft tissue infection of the lower limb and compares the results with the existing literature.
The Introduction section provides background related to the epidemiology of DFU, NPWT, and its impact on the patient's outcome. Moreover, the mechanisms that explain this relationship are explained in a short paragraph.
The results section presents the main characteristics of patients with DFU treated with NPWT, showing considerable variability in terms of DFU etiology and severity. This variability is relevant, as it may interfere with the reliability of the results and the ability to draw firm conclusions regarding the role of NPWT in DFU treatment.
The discussion cites results from different meta-analyses without specifying key wound characteristics (severity, type, location, size, bacterial load), which are crucial factors influencing DFU outcomes under NPWT. Furthermore, the authors must contextualize their findings against the evidence from meta-analyses and current clinical guidelines to substantiate their conclusions.
Author Response
Dear Reviewer 2,
First and foremost, we would like to express our sincere gratitude for your careful review of our manuscript entitled “Negative Pressure Wound Therapy in the Treatment of Complicated Wounds of the Foot and Lower Limb in Diabetic Patients.” Your thoughtful observations, insightful critiques, and valuable references have significantly contributed to the improvement of our work. We truly appreciate your engagement with the topic and the constructive tone of your feedback.
We have carefully addressed each of your comments and implemented the necessary clarifications and changes to the manuscript. Please find below our point‑by‑point responses:
Comment 1: The results section presents the main characteristics of patients with DFU treated with NPWT, showing considerable variability in terms of DFU etiology and severity. This variability is relevant, as it may interfere with the reliability of the results and the ability to draw firm conclusions regarding the role of NPWT in DFU treatment.
We have expanded the results section and reconfigured the tables and figures included to provide a better understanding of our cohort. Information on additional comorbidities, types of lesions, germs isolated, outcomes and therapy used were incorporated. The variability of the lesions was stated further in the Discussion section and an interpretation on how it affects the generalization of results was provided.
Comment 2: the authors must contextualize their findings against the evidence from meta-analyses and current clinical guidelines to substantiate their conclusions.
The Discussion section was heavily redacted and expanded as per your suggestions. Comparative analysis of available literature with our study is included. A comprehensive Limitations subsection was added at the end of the Discussion. It addresses: retrospective design, small sample size, clinical heterogeneity, information bias and lack of long-term follow-up, non-generalizability due to local clinical practice. The discussion explicitly acknowledges that causality cannot be established due to the observational nature of the study and emphasizes the need for individualized treatment decisions.
Once again, we are very grateful for your careful reading and detailed suggestions. We believe the manuscript is now substantially improved in both content and clarity. We remain open to any additional suggestions you may have.
With sincere appreciation,
The Authors
Reviewer 3 Report
Comments and Suggestions for Authors
INTRODUCTION
- Can you clarify and refine the opening definition of diabetic foot disease to improve readability and precision?
- While the burden of diabetic foot is well covered, the introduction would benefit from a smoother transition to NPWT — can you expand a bit on why conventional therapies are insufficient, leading to NPWT adoption?
PATIENTS AND METHODS
- Could you specify the exact study period (start and end dates) and total number of patients included?
- What were your exclusion criteria (e.g. patients with ischemic gangrene, severe PAD, or non-diabetic ulcers)?
- Can you provide details on the NPWT protocol, device/brand used, negative pressure settings (mmHg). Whether continuous or intermittent therapy was applied?
- Why did you define the outcome only as “granulated wound vs major amputation”? Could additional clinically relevant outcomes (e.g., wound healing time, infection resolution, limb salvage rates) be included?.
RESULST
- Could you clarify why only 30 patients were included over 15 years? Were patients excluded, or was NPWT rarely used in your center?
- You state there was “no association” between outcomes and WIFI/IWGDF/TPI classifications, what statistical test did you use to evaluate this?
- Why was the outcome defined mainly as “granulation vs amputation”? Could you provide secondary outcomes such as healing time, length of hospital stay, re-intervention rates, or long-term limb salvage?
DISCUSSION
- You conclude that NPWT avoided amputation in 80% of cases. How do you reconcile this with your finding that only 6 patients achieved full healing at discharge?
- How do your results compare directly with the RCTs you cite (e.g., Armstrong, Blume, DiaFu)?
- Since the IWGDF 2023 guidelines advise against NPWT for infected wounds, how do you justify its use in your 24 septic cases?
- Could you expand on why your study found no correlation between outcomes and classification systems (WIFI, IWGDF, TPI), while others emphasize their predictive value?
- What are the main limitations of your study (sample size, retrospective design, heterogeneity, lack of control group), and how do these affect interpretation of your findings?
CONCLUSIONS
- Given your small and heterogeneous sample, do you think it is appropriate to state that NPWT “remains a very useful tool”? Could you reframe this more cautiously?
- How do you reconcile your conclusions with the 2023 IWGDF guidelines, which do not recommend NPWT in infected DFUs?
- Would you consider explicitly acknowledging the limitations of your study in the conclusion (sample size, retrospective design, lack of control group)?
- Instead of predicting that RCTs will “prove beyond doubt,” could you suggest that further high-quality RCTs are needed to clarify NPWT’s role?
-
The manuscript needs language editing before submission.
-
A professional English editing service or a native speaker with medical writing experience should polish it.
-
The main focus should be on:
-
Correct medical terminology (e.g., “etiological,” not “ethological”).
-
Grammar and sentence flow.
-
Removing speculative or overly strong statements.
-
Keeping the tone scientific and cautious.
-
Author Response
Dear Reviewer 3
We would like to express our sincere gratitude to Reviewer 1 for the thorough and constructive review of our manuscript entitled: "Negative pressure wound therapy in the treatment of complicated wounds of the foot and lower limb in diabetic patients: a retrospective case series and literature review".
We deeply appreciate the reviewer’s insightful comments and helpful suggestions, which allowed us to substantially improve the clarity, structure, and scientific rigor of the manuscript. We have carefully revised the manuscript according to all the points raised and provide below a detailed, point-by-point response. All modifications were incorporated into the revised version and highlighted accordingly.
Comment 1: Can you clarify and refine the opening definition of diabetic foot disease to improve readability and precision? While the burden of diabetic foot is well covered, the introduction would benefit from a smoother transition to NPWT — can you expand a bit on why conventional therapies are insufficient, leading to NPWT adoption?
The entire Introduction section was completely rewritten and all suggestions were incorporated.
Comment 2: Could you specify the exact study period (start and end dates) and total number of patients included? What were your exclusion criteria (e.g. patients with ischemic gangrene, severe PAD, or non-diabetic ulcers)? Can you provide details on the NPWT protocol, device/brand used, negative pressure settings (mmHg). Whether continuous or intermittent therapy was applied?
All suggested modifications and additional information are now incorporated into the Methods section.
Comment 3: Why did you define the outcome only as “granulated wound vs major amputation”? Could additional clinically relevant outcomes (e.g., wound healing time, infection resolution, limb salvage rates) be included?.
The explanation is now provided. Unfortunately, due to the retrospective nature of the study and heterogeneity of follow-up some details were unavailable to us (e.g. total healing time, overall limb salvage rate). We could only provide the actual medical observations on discharge of patients (and that is observable healing in grated patients or granulated non-infected wounds in the rest of patients which were scheduled for additional consults and ulterior therapy). We added additional available information on patient evolution in the Result section.
Comment 4: Could you clarify why only 30 patients were included over 15 years? Were patients excluded, or was NPWT rarely used in your center?
The explanation is now clearly stated in the Methods section. The 30 patients represent the totality of consecutive patients meeting inclusion criteria treated by the same team with NPWT.
Comment 5: You state there was “no association” between outcomes and WIFI/IWGDF/TPI classifications, what statistical test did you use to evaluate this?
Details on the statistical analysis is now included in the Methods section and a new table 2 provides actual statistical results.
Comment 6: You conclude that NPWT avoided amputation in 80% of cases. How do you reconcile this with your finding that only 6 patients achieved full healing at discharge?
As stated before patients were discharged with non-infected granulated wounds on their way to complete healing and with scheduled consultations for further follow-up and eventual treatment. No necrosis was present and the remaining limb was viable at the time of patient discharge,
Comment 7: Since the IWGDF 2023 guidelines advise against NPWT for infected wounds, how do you justify its use in your 24 septic cases?
A comprehensive explanation for usage of NPWT in septic cases was provided in Introduction section.
Comment 9: How do your results compare directly with the RCTs you cite (e.g., Armstrong, Blume, DiaFu)? What are the main limitations of your study (sample size, retrospective design, heterogeneity, lack of control group), and how do these affect interpretation of your findings?
The Discussion section was heavily redacted and expanded as per your suggestions. Comparative analysis of available literature with our study is included. A comprehensive Limitations subsection was added at the end of the Discussion. It addresses: retrospective design, small sample size, clinical heterogeneity, information bias and lack of long-term follow-up, non-generalizability due to local clinical practice. The discussion explicitly acknowledges that causality cannot be established due to the observational nature of the study and emphasizes the need for individualized treatment decisions
Comment 10: Given your small and heterogeneous sample, do you think it is appropriate to state that NPWT “remains a very useful tool”? Could you reframe this more cautiously? Instead of predicting that RCTs will “prove beyond doubt,” could you suggest that further high-quality RCTs are needed to clarify NPWT’s role?
We have reformulated into a more reasonable conclusion
Comment 11. Quality of English language
The manuscript has been extensively revised for grammar, clarity, and academic tone by an English-speaking editor. We have aimed to maintain a fluent and formal style throughout.
We hope that these extensive revisions have addressed all the points raised and improved the clarity and scientific quality of our work. We are grateful for the opportunity to revise our manuscript and look forward to your further consideration.
Respectfully,
The Authors
Round 2
Reviewer 1 Report
Comments and Suggestions for Authors
I appreciate the authors' efforts to address my concerns and update their manuscript. Although my opinion about the depth of the study remains the same, I am now satisfied with the changes made excpt for the references section that was not improved as suggested:
Comment 7. The background on diabetes and NPWT is appropriate. There is a gap regarding the study’s rationale and specific objectives are stated but not sufficiently explicit. Clearly define the study’s primary outcome (major amputation avoided) and secondary outcomes (wound healing, complications, mortality). Cites prior research effectively to highlight the gap in literature. Please, regarding the use of negative pressure for the treatment of diabetic foot ulcers and Charcot foot ulcerations, add a few lines and quote: https://pubmed.ncbi.nlm.nih.gov/36096551/
Please, provide it.
Author Response
We sincerely thank Reviewer 1 for their continuous involvement with our manuscript and for the constructive feedback provided throughout the revision process.
We are particularly grateful for the most recent suggestion regarding the clarification of the study objectives and the addition of relevant literature. As requested, we have revised the Introduction to more clearly state the primary outcome (major amputation avoided) and secondary outcomes (wound healing, complications, mortality).
Additionally, we have incorporated the reference suggested by the reviewer regarding the use of negative pressure therapy in diabetic foot ulcers and Charcot foot ulcerations. This citation is now included in the manuscript as Reference 5, and the corresponding discussion has been added in the Introduction section.
We appreciate the opportunity to further improve our work and we hope that the current version of the manuscript meets the expectations.
With kind regards,
The Authors